# Homophone Disambiguation Reveals Patterns of Context Mixing in Speech Transformers

**Hosein Mohebbi**[1]  **Grzegorz Chrupała**[1]  **Willem Zuidema**[2]  **Afra Alishahi**[1]

[1] CSAI, Tilburg University  [2] ILLC, University of Amsterdam

{h.mohebbi, a.alishahi}@tilburguniversity.edu

grzegorz@chrupala.me

w.h.zuidema@uva.nl

## Abstract

Transformers have become a key architecture in speech processing, but our understanding of how they build up representations of acoustic and linguistic structure is limited. In this study, we address this gap by investigating how measures of 'context-mixing' developed for text models can be adapted and applied to models of spoken language. We identify a linguistic phenomenon that is ideal for such a case study: homophony in French (e.g. *livre* vs *livres*), where a speech recognition model has to attend to *syntactic cues* such as determiners and pronouns in order to disambiguate spoken words with identical pronunciations and transcribe them while respecting grammatical agreement. We perform a series of controlled experiments and probing analyses on Transformer-based speech models. Our findings reveal that representations in encoder-only models effectively incorporate these cues to identify the correct transcription, whereas encoders in encoder-decoder models mainly relegate the task of capturing contextual dependencies to decoder modules.[1]

## 1 Introduction

In both speech and text processing, variants of the Transformer architecture (Vaswani et al., 2017) have become ubiquitous. The key advantage of this neural network topology lies in the modeling of pairwise relations between elements of the input (tokens): the representation of a token at a particular Transformer layer is a function of the weighted sum of the transformed representations of all the tokens in the previous layer. This feature of Transformers is known as *context mixing* and understanding how it functions in specific model layers is crucial for tracing the overall information flow. Several works (e.g. Kobayashi et al., 2020, 2021; Mohebbi et al., 2023) have proposed methods to quantify context mixing and applied them

to transformers trained on text data. In the current study we investigate context mixing in models trained on spoken language data.

We focus on the task of automatic speech recognition (ASR), which consists of transcribing spoken utterances into their written equivalent. In order to probe context mixing for speech models, we exploit a quirk of the French spelling system: modern spoken French does **not** in general[2] mark grammatical number on nouns, adjectives or third-person verbs, only on determiners and pronouns – however written French reflects an earlier period of the language and **does** mark nouns, adjectives and verbs for number. Therefore, for an ASR system to produce grammatical transcriptions which feature number agreement, the model needs to rely on very specific cues in the input and/or in the prefix of the generated output.

For example, when transcribing the utterance *Elle a perdu les **livres***, meaning "She lost the books", when generating the bolded TARGET word, the model needs to decide whether to output *livre* (singular) or rather *livres* (plural) and it cannot use the speech signal corresponding to the target word to choose the correct form, as in spoken French both are pronounced /livʁ/. The correct choice is determined by grammatical agreement with the underlined CUE word, here the plural determiner, which is different from the singular form, both in speech (/le/ vs /lə/) and in writing (*les* vs *le*).

Thanks to this phenomenon we can form a strongly supported prior expectation that the representation of the cue token should contribute to the representations of the target. We are therefore able to validate previously suggested context mixing scoring methods on a dataset of French utterances featuring such grammatical homonyms: we find that context-mixing scoring methods, and especially Value Zeroing, behave plausibly, largely

---

[1] Code is freely available at https://github.com/hmohebbi/ContextMixingASR

[2] There is a small number of irregular exceptions to this phenomenon.

agreeing with prior expectations, as well as with targeted diagnostic probes and faithfulness tests.

We furthermore apply context-mixing scoring methods to two types of architectures: encoder-decoder and encoder-only, and discover interesting differences in how they behave. In encoder-only models, spoken cue-word representations contribute to target representations in a straightforward manner. Meanwhile, in encoder-decoder architectures it is mostly the decoder which ensures grammatical number agreement, and therefore spoken cue-word representations contribute little to targets, while written cue-words in the generated prefix make a decisive contribution.

## 2 Related Work

Our work builds on two strands of research: one focusing on analyzing models of spoken language, and the other focusing specifically on quantifying information flow due to context-mixing in Transformer models of textual data.

### 2.1 Analyses of speech models

A large body of work on analyzing Transformer-based speech models has focused on probing their representations to uncover various types of linguistic structures they are hypothesized to encode, such as phonemic, lexical, syntactic information, as well as non-linguistic information such as speaker identity and demographics (Pasad et al., 2021; Shah et al., 2021; Millet and Dunbar, 2022; Yang et al., 2023; Shen et al., 2023). Pasad et al. (2021) showed that in pre-trained wav2vec 2.0 models (Baevski et al., 2020), the representations evolve layer by layer according to an acoustic-linguistic hierarchy: the shallowest layers primarily encode acoustic features, followed by layers encoding phonetic, lexical, and semantic information. This trend then reverses in the deeper layers, unless the model is fine-tuned for ASR. More recently, Shen et al. (2023) showed that pre-trained wav2vec 2.0 models capture some degree of syntactic structure, particularly in the middle layers of the network.

Another strand of research is focused on analyzing self-attention weights inside a Transformer layer as a simple measure of pairwise interaction of frame representations. By analyzing self-attention weights in Conformer (Gulati et al., 2020), Shim et al. (2022) observe two distinct patterns for phonetic and linguistic localization in the lower and upper layers, respectively. Audhkhasi et al. (2022)

measure the diversity of attention heads in Conformer and find that the attention probabilities from different heads show low diversity (analogous to the redundant pattern of attention in pre-trained language models) during the course of training, implying that they mostly focus on the same frames. In contrast, the value vectors show high diversity among different heads.

### 2.2 Analyses of context mixing

While self-attention weights were initially deemed as the primary source of information mixing in Transformers (Clark et al., 2019; Kovaleva et al., 2019; Reif et al., 2019; Lin et al., 2019), recent studies have shown that integrating other components in the architecture can yield more reliable measures (Kobayashi et al., 2020, 2021; Modarressi et al., 2022; Ferrando et al., 2022; Mohebbi et al., 2023). Kobayashi et al. (2020) proposed a context-mixing technique called Attention Norm which is based on the norm of the transformed value vectors, and demonstrated that the model may assign a high attention weight to a token with a small norm, particularly for highly frequent and less informative words. More recently, Mohebbi et al. (2023) introduced a technique called Value Zeroing, which offers more reliable context mixing predictions compared to alternatives, mainly due to incorporating the output representations of Transformer layers that encompass all the components within the layer, including residual connections and feed-forward networks.

These techniques are developed for and evaluated on text. In this study we borrow the state-of-the-art context mixing methods, Attention Norms and Value Zeroing, and extend their applicability to both encoder-only and encoder-decoder-based speech models.

## 3 Context Mixing

Our goal is to quantify the contribution of a cue word to the representation of a target word across layers of speech Transformers. Here we first review the formation of contextualized representations in these models, and then describe how we adapt text-based context mixing methods for our purpose.

### 3.1 Background

**Transformer Encoder.** A speech encoder takes as input a sequence of audio representations known as audio *tokens*, or *frames*, and passes them through

a stack of encoders to make contextualized audio representations. These tokens are typically the output of a convolutional feature extractor module applied to the acoustic waveform. Given a sequence of $d$-dimensional input audio representations $(\boldsymbol{x}_1, ..., \boldsymbol{x}_T)$, the goal of a Transformer encoder layer is to build contextualized audio representations $(\tilde{\boldsymbol{x}}_1, ..., \tilde{\boldsymbol{x}}_T)$ for each frame in the context. This process consists of two sublayers: a multi-head self-attention mechanism (MHA) and a position-wise fully connected feed-forward network (FFN).

After passing through a layer normalization (LN)[3], each input vector in MHA module is compressed into separate query $(\boldsymbol{q}_i^h)$, key $(\boldsymbol{k}_i^h)$, and value $(\boldsymbol{v}_i^h)$ vectors via trainable linear transformations for each head $h \in \{1, ..., H\}$:

$$\boldsymbol{q}_i^h = \text{LN}_{\text{MHA}}(\boldsymbol{x}_i)\boldsymbol{W_Q^h} + \boldsymbol{b}_Q^h \qquad (1)$$

$$\boldsymbol{k}_i^h = \text{LN}_{\text{MHA}}(\boldsymbol{x}_i)\boldsymbol{W_K^h} + \boldsymbol{b}_K^h \qquad (2)$$

$$\boldsymbol{v}_i^h = \text{LN}_{\text{MHA}}(\boldsymbol{x}_i)\boldsymbol{W_V^h} + \boldsymbol{b}_V^h \qquad (3)$$

The raw attention weight $\alpha_{i,j}$, representing how much the $i^{\text{th}}$ frame attends to the $j^{\text{th}}$ frame within the context, is then computed by scaled dot-product between the corresponding query and key vectors followed by the softmax function:

$$\alpha_{i,j} = \text{softmax}\left(\frac{\boldsymbol{q}_i \boldsymbol{k}_j^\top}{\sqrt{d_h}}\right) \in \mathbb{R} \qquad (4)$$

Consequently, the context vector output from MHA can be formulated as a weighted sum over the transformed value vectors within all frames across attention heads:

$$\boldsymbol{z}_i = \sum_{h=1}^{H} \sum_{j=1}^{T} \alpha_{i,j}^h \boldsymbol{v}_j^h \mathbf{W}_O^h + \boldsymbol{b}_O + \boldsymbol{x}_i \qquad (5)$$

Finally, two linear transformations with a GELU activation function (Hendrycks and Gimpel, 2016) in between are applied to every $\boldsymbol{z}_i$ to produce output frame representation $\tilde{\boldsymbol{x}}_i$:

$$\tilde{\boldsymbol{x}}_i = \text{GELU}\left(\text{LN}_{\text{FFN}}(\boldsymbol{z}_i)\mathbf{W}_1 + \boldsymbol{b}_1\right)\mathbf{W}_2 + \boldsymbol{b}_2 + \boldsymbol{z}_i \qquad (6)$$

_______

[3]As apposed to the original Transformer architecture (Vaswani et al., 2017), models studied in this paper utilize pre-LN rather than post-LN.

**Transformer Decoder.** In encoder-decoder-based ASR models, the model receives textual tokens as input to generate a probability distribution over the target vocabulary. The decoder then constructs contextualized representations based on both the previously generated text tokens from earlier time steps and the final encoded audio representation. The procedure is similar to encoders, except they perform the MHA procedure (Eq. 5) twice in two sublayers. The first MHA sublayer is modified to prevent tokens attending to subsequent positions when forming context vectors. The second MHA sublayer uses the key and value vectors from the last encoder layer to incorporate encoded audio representations into computation through cross-attention.

**Decoder-less ASR.** For the encoder-only architecture, ASR is carried out by outputting transcription tokens directly via a connectionist temporal classification (CTC) layer on top of the encoder output.

### 3.2 Analysis methods

In an encoder-decoder model, we compute three types of context mixing scores: within-encoder, within-decoder and cross (between encoder and decoder).

**Within-encoder scores** measure the contribution of audio frame representations to a target frame representation in the encoder.

**Within-decoder scores** measure the contribution of token representations from previous time steps to a target token in the decoder.

**Cross scores** measure the contribution of audio frame representations from the encoder to a target token representation in the decoder.

To compute within-encoder and cross scores for audio input, we need to extract frames that correspond to each cue or target word (based on the alignment between the audio and the transcription). Consider sets $\mathcal{I}$ and $\mathcal{J}$, representing the frame indices for words $i$ and $j$, respectively. These sets allow us to aggregate word-level context mixing scores $(S_{i \leftarrow j})$ to measure how much the $j^{\text{th}}$ word in the context contributes to the representation of the $i^{\text{th}}$ word. In the decoder, these sets contain all the subword tokens corresponding the word of interest, and thus often contain just a single value.

We describe each context mixing method below.

**Attention (Attn).** Raw attention weights are derived from Eq. 4, averaged over all frames for each of the two words, over all heads:

$$S_{i \leftarrow j} = \frac{1}{|\mathcal{I}||\mathcal{J}|H} \sum_{n \in \mathcal{I}} \sum_{m \in \mathcal{J}} \sum_{h=1}^{H} \alpha_{n,m}^h \quad (7)$$

**Attention Norm (AN).** Motivated by Eq. 5, this metric measures the Euclidean norm of the weighted transformed vector to measure each token's contribution in a self-attention module (Kobayashi et al., 2020):

$$S_{i \leftarrow j} = \frac{1}{|\mathcal{I}||\mathcal{J}|H} \sum_{n \in \mathcal{I}} \sum_{m \in \mathcal{J}} \sum_{h=1}^{H} \|\alpha_{n,m}^h \boldsymbol{v}_m^h \mathbf{W}_O^h\| \quad (8)$$

**Value Zeroing (VZ).** It measures how much the output representation of token $i$ is affected when excluding the $j^{\text{th}}$ token by zeroing its value vector (Mohebbi et al., 2023). In the speech context, we set the value vectors in all frames in $\mathcal{J}$ (corresponding to word $j$) to zero, extract the alternative representations for each frame $n \in \mathcal{I}$ (that is, $\tilde{\boldsymbol{x}}_n^{\neg j}$), and measure how much each frame representation has changed compared to the original ones:

$$S_{i \leftarrow j} = \frac{1}{|\mathcal{I}|} \sum_{n \in \mathcal{I}} \cos(\tilde{\boldsymbol{x}}_n, \tilde{\boldsymbol{x}}_n^{\neg j}) \quad (9)$$

## 4 Experimental Setup

### 4.1 Data

As a case study for analyzing word-level context mixing in audio representations, we opt for homophony in French, where certain grammatical forms of a word are identical in pronunciation, despite having different spellings. The spoken forms are thus disambiguated in writing by relying on grammatical agreement phenomena, where nouns agree in number with determiners, and verbs agree with subject pronouns. The nature of this task makes it suitable for analyzing context mixing in speech models since it offers a strong hypothesis regarding which part of the context is most relevant when transcribing ambiguous target words.

We define three specific syntactic templates in which homophony may appear. All rely on grammatical number agreement: the first one captures determiner-noun number agreement; the second – number agreement between subject pronoun and verb; and the third – number agreement between

subject noun phrase and verb. Table 1 shows examples of utterances captured using these templates.

We used the test set of the Common Voice (Ardila et al., 2020) spoken corpus[4], which consists of 26 hours of transcribed speech in French, and generated dependency trees of transcriptions with SpaCy (Honnibal and Montani, 2017) to discover instances of these templates. We selected those examples for which all models in our analysis predict the cue and target words correctly. This resulted in the extraction of 1000 audio-transcription pairs.[5]

### 4.2 Target models

Our experiments cover five prominent off-the-shelf models based on Transformers for learning speech representations, which can be classified into two categories in terms of their architecture and training objectives. See Table 2 for an overview.

**Whisper family.** Whisper (Radford et al., 2022) is a multilingual encoder-decoder multitask speech model. It takes raw audio as input, splits it into 30-second chunks, and transforms them into log-Mel spectrograms. After passing through two convolution layers, these audio features are then fed into a stack of Transformer encoder layers to create contextualized audio representations. The model is autoregressively trained to predict next token on a set of supervised audio-to-text tasks such as language identification, phrase-level timestamps, multilingual speech transcription, and speech translation to English. For our experiments, we used Whisper in three different sizes (base, small, medium).

**Wav2vec2 family.** This family of models employs only the encoder part of the Transformer architecture, built on the wav2vec 2.0 framework (Baevski et al., 2020). The model receives raw audio waveform input and maps it into latent speech representations via a multi-layer convolutional neural network. These latent representations are partially masked (Jiang et al., 2019; Wang et al., 2020) and passed through a stack of Transformer encoder layers to build contextualized representations. The model is then trained by solving a contrastive task over quantized speech representations in a self-supervised manner. After pre-training, the model can be fine-tuned on labeled data using a Connec-

---

[4]We only used the test split of the corpus since the training and validation sets have already been utilized during the fine-tuning phase of models within the Wav2vec2 family.

[5]The details of the data extraction procedure is described in Appendix A.

| Pattern | Examples of transcription | # |
|---------|---------------------------|---|
| Det_Noun | C'est le septième **titre** de champion de Syrie de l'histoire du club
Il y mène une **vie** d'études et de recherches | 720 |
| Pronoun_Verb | Chaque jour, leurs concurrents les voient sortir de pistes dont ils **ignorent** l'existence
On y **trouve** une plage naturiste | 257 |
| Det_Noun_Verb | Peu après cette élimination, le **club** et Alexander se **séparent** à l'amiable
À la fin, les **enfants** se **révoltent** et détruisent l'école. | 23 |

Table 1: Examples of the extracted audios from the Common Voice corpus based on defined patterns. Last column shows the number of examples obtained. Cue and Target words are underlined and **bolded**, respectively.

| Model | Architecture | Size | Layers | Width | Pre-training | Multilingual | Fine-tuned |
|-------|--------------|------|--------|-------|--------------|--------------|------------|
| Whisper | encoder-decoder | base
small
medium | 6
12
24 | 512
768
1024 | supervised | yes | no |
| XLSR-53
XLSR-1 | encoder | large | 24 | 1024 | self-supervised | yes
no | yes
yes |

Table 2: Specification of the target models in our study.

tionist Temporal Classification (CTC) loss (Graves et al., 2006) for downstream speech recognition. For our experiments, we used XLSR-53[6] (Conneau et al., 2020), and XLSR-1[7] (Evain et al., 2021) from this family. The former is pre-trained on 56K hours of speech data in 53 languages, whereas the latter is pre-trained on 7K hours of French speech only. Both models are fine-tuned for French ASR using CTC.

### 4.3 Frame-word alignment

To investigate context mixing in speech models at the word level, we need to know which frames in the encoder representation correspond to which words in the audio. As the Common Voice dataset does not provide this information, we used Montreal Forced Aligner (McAuliffe et al., 2017, MFA) to extract the start ($t_s$) and end time ($t_e$) of each word in an utterance, and mapped them to boundary frames $f_s$ and $f_e$:

$$f = \lceil \frac{t}{\mathcal{T}} \times T \rceil \tag{10}$$

where $\mathcal{T}$ and $T$ denote the total time of a given utterance and the total number of frames in the encoded audio, respectively.[8] The range ($f_s$, $f_e$) indicates which frames in the encoder correspond

to the selected word. In our dataset, on average a word corresponds to 369ms of audio duration and 19 encoder frames.

## 5 Experiments

We design a number of experiments meant to first validate the context-mixing scoring methods in the context of speech transformers, and further to provide converging evidence on the specific patterns of context-mixing in the two target architectures on our dataset.

### 5.1 Cue contribution

To quantify the contribution of the cue word to the representation of a target word, we compute context mixing maps based on different methods described in Sec. 3.2. For each method, we normalize the scores for all words in a sentence so that they are positive values and sum up to one. We then select the $t^{\text{th}}$ row of each context mixing map, where $t$ represents the time step at which a target word is being generated. The resulting 1-D array indicates how much the target word relies on other words in the context when forming its contextualized representation.

Recall that all models in our experiments accurately transcribe both cue and target words, as we

[6]https://huggingface.co/jonatasgrosman/wav2vec2-large-xlsr-53-french
[7]https://huggingface.co/LeBenchmark/wav2vec2-FR-7K-large
[8]In contrast to the models in the Wave2Vec 2.0 family, Whisper does not support attention mask in its encoder and

instead pads the audio input with silence to a constant duration of 30 seconds in its processor. Therefore the variables $\mathcal{T}$ and $T$ in Eq. 10 are always set as 30 seconds and 1500, respectively for Whisper models.

specifically extracted the dataset under this condition. As cue words are the sole indicators in the context for identifying the correct transcription of target words, we expect that accurate disambiguation of the target word is due to the model's reliance on the cue words when constructing target word representations. To measure this reliance, we define a binary cue vector $C$ according to the following condition:

$$C_i = \begin{cases} 1, & \text{the } i^{\text{th}} \text{ position} \in \text{Cue word} \\ 0, & \text{otherwise} \end{cases} \quad (11)$$

We then compute the *Cue contribution* score as the dot product of the cue vector ($C$) and a given context mixing score ($S_t$).

**Results.** Figure 1 shows cue contribution scores for all layers of both randomly initialized and trained versions of XLSR-53, which is an encoder-based model. For the model with random weights, the cue contribution scores are low and uniform across the layers. For the trained model, cue contribution scores indicate that cues are prominently integrated into the target audio representations in the middle layers (notably, at layers 10, 12 and 15). This pattern conforms to the prior expectations for the grammatical number homonymy task.

Figure 2 displays the cue contribution scores for the Whisper-medium model. Results show that the within-encoder scores are not noticeably higher than the ones for the randomly initialized model, except for layers 19–21 and 23–24 which show a slight increase. In contrast, the cross and within-decoder scores show that the decoder prominently incorporates the cue word into its contextualized representation of the target word.

In the decoder, each layer has two options for information integration (cf. Eq. 5): attending to previously generated tokens or attending to the audio representations output from the last encoder layer. By comparing the cross and within-decoder alignment scores, we find that the representations in the decoder strongly capture the syntactic dependency on the cue word through the self-attention module. We conjecture that in the encoder-decoder model, the decoder plays a crucial role in capturing syntactic dependency and does not rely on cue word representations in the encoder, probably due to having access to text-token representations as an easier alternative. We examine the role of the cross-attention module more carefully in Section 5.3.

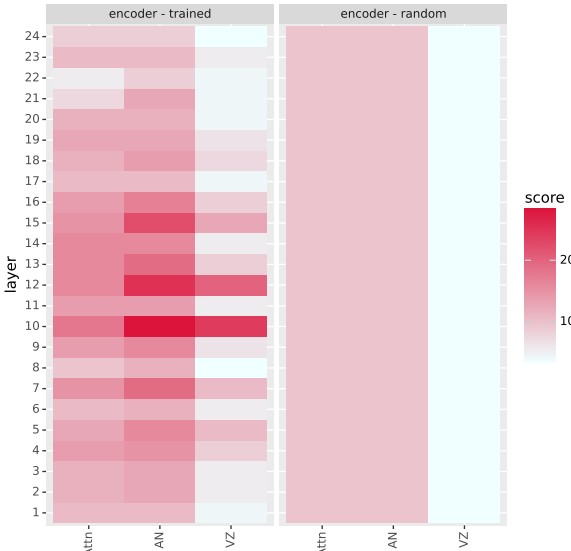

Figure 1: Layer-wise cue contribution according to different analysis methods averaged over all examples for XLSR-53, trained (left) vs. randomly initialized (right).

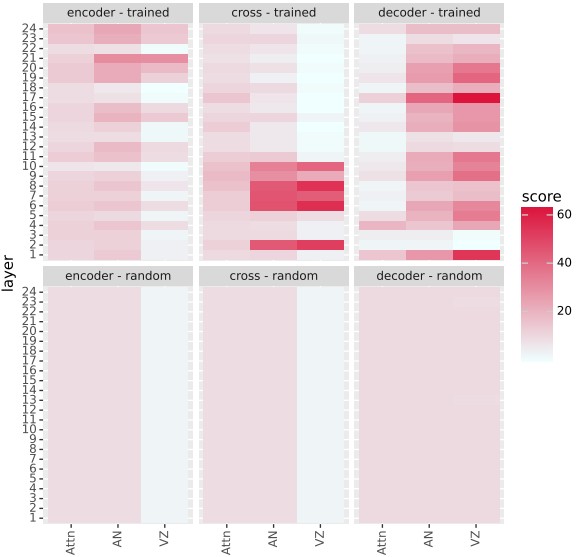

Figure 2: Layer-wise cue contribution according to different analysis methods averaged over all examples for Whisper-medium, trained (top) vs. randomly initialized (bottom).

For all these scores, predictions made by Value Zeroing and Attention Norms methods are largely consistent with each other, with Value Zeroing suggesting a stronger reliance on cue word in the decoder of the trained model. In contrast, raw attention scores are uniform and less informative, as shown also in previous studies (Kobayashi et al., 2020; Hassid et al., 2022; Mohebbi et al., 2023).

Figure 3 shows the context-mixing map for

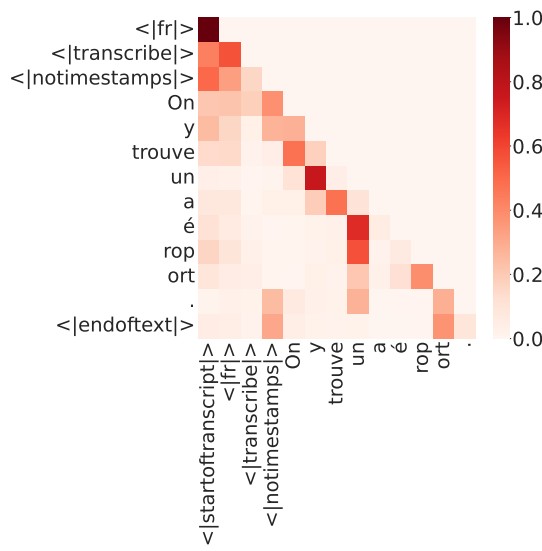

Figure 3: **Within-decoder** Value Zeroing context mixing map for Whisper-medium at layer 10 for our running example. Target word 'trouve' representations notably rely on the cue word 'On' in the context. Similarly, the cue word 'un' notably contributes to forming representations of the word 'aéroport'.

layer 10 for an example from the dataset: *"On y trouve un aéroport.*", illustrating the dependence of target subword tokens on the cue word in the decoder prefix.

### 5.2 Probing number encoding

Since the main source of information for disambiguating the target word is its number agreement with the cue word, we hypothesize that layers with a higher cue contribution score must encode number information more strongly and/or accessibly. To test our hypothesis, we conduct a probing experiment by extracting target word representations across all layers, and associating each representation with a *Singular* or *Plural* label. We then train a Logistic Regression classifier with L2 regularization on these representations to predict target grammatical number.

**Results.** Figure 4 presents the 3-fold cross-validation accuracy of the probing classifiers for different models across all layers. As a baseline, we also include the results for target representations obtained from randomly initialized models.

For Whisper decoders, probing accuracy is close to one – indeed the accuracy is very high even for randomly initialized target models suggesting that simple word identity is being used by the diagnostic

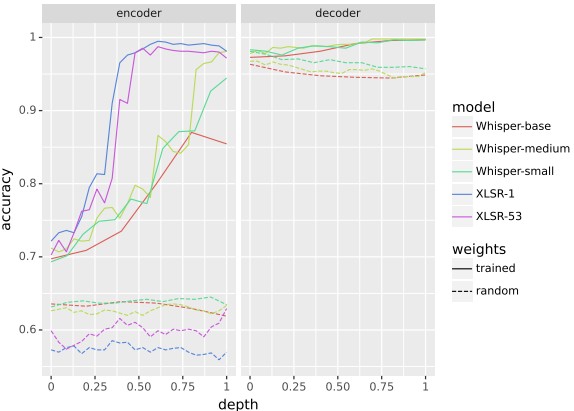

Figure 4: Accuracy of probing classifiers trained on frozen target representations obtained from various ASR models. The depth of Whisper-base (6) and Whisper-small (12), has been normalized to 1 to facilitate comparisons.

classifier to predict grammatical number.

Encoder representations in encoder-based and encoder-decoder-based models, however, display two distinct patterns. Whisper encoders show a gradual increase as we progress through the layers. The encoder of Whisper-medium shows a jump at layer 20, the same layer for which we observed a significant cue contribution score in Section 5.1. Comparing Whisper models with different sizes, we can see that the gap between encoder and decoder decreases as the model size increases. In contrast, XLSR-53 and XLSR-1 display a noticeable spike in the middle layers (notably at layer 10), the same layers for which we observed higher cue contribution scores. This implies that the target words in the middle layers of these models significantly incorporate the cue words in their representations and encode number. This information can be transferred to the final layer with minimal loss, where the model is able to correctly disambiguate the target word.

### 5.3 Silencing the cue words

Results of the previous experiments in Section 5.2 suggest that in encoder-decoder models, target representations in the decoder show a stronger encoding of syntactic dependency on the cue word compared to encoder representations. Additionally, our analysis of the context mixing scores in Section 5.1 revealed that encoder-decoder models exploit the agreement information provided by the cue word via its text representation in the decoder, rather than relying on the encoded spoken cue representation

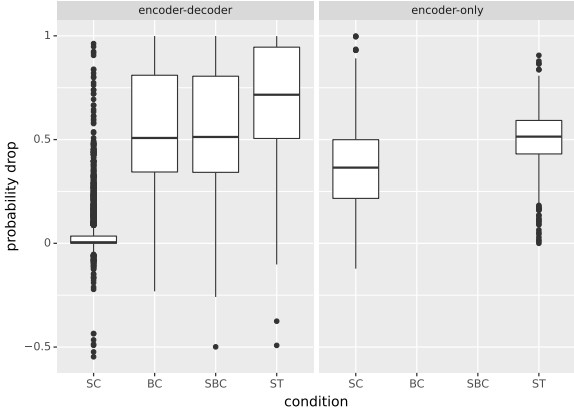

Figure 5: Changes in the models' confidence in generating the target word in different setups of input ablations, averaged over models within each group. SC=Silence cue, BC=Blank cue, SBC=Silence & blank cue, ST=Silence target.

via the cross-attention module.

In this section, we employ the concept of input ablation (Covert et al., 2020) to assess the influence of cue words on an ASR model's decision: we measure how much the model's confidence drops when the cue word is excluded from the input during the transcription of ambiguous target words. For encoder models, we silence the frames corresponding to the cue word in the raw audio and provide the modified audio input to the model. For encoder-decoder models, we consider the following conditions: silencing the cue words from the raw audio input (silence cue), replacing the cue token with an unknown token in the input of the decoder (blank cue), and combining both modifications simultaneously (silence & blank cue). As a control we also include the condition where we silence the target word in the input to the encoder (silence target) – in this case we expect the models' confidence to drop irrespective of architecture, as this manipulation removes most of the information available to the model about the identity of the target word.

We compute the drop in model's output probability as $p(y_t|\boldsymbol{e}) - p(y_t|\boldsymbol{e}\backslash e_t)$, where $y_t$ denotes the logit value corresponding to the target word at time step $t$, and $\boldsymbol{e}$ refers to a given raw input audio at encoder setup, and original previously generated tokens at decoder setup.

**Results.** Figure 5 presents the results for different ASR models, averaged over each model family. Notably, in encoder-decoder models (models in the Whisper family), we can see that silencing the cue

word in the encoder input does not significantly affect the model's confidence when generating the target word. However, if we blank out the cue token in the decoder input while keeping the encoder input intact, the model experiences a significant decrease in confidence in its target word prediction. The same drop can also be observed when both the encoder and decoder inputs have the cue words silenced/blanked out. These findings support our hypothesis that in encoder-decoder models, the decoder captures the syntactic dependency on the cue words in the target representations by attending to the cue word from its prefix rather than relying on the encoded cue words from the encoder. The scenario for encoder-based models, however, is different. As we can see, silencing the cue words in the encoder input leads to a significant drop in the model's confidence in its target word prediction. This corroborates with the results observed in Sections 5.1 and 5.2.

# 6 Conclusion

In this paper, we adapted state-of-the-art context mixing methods from the text domain to analyze sensitivity to syntactic cues and to trace information mixing in Transformer-based models of speech recognition. Our analyses revealed that in encoder-only models, spoken cue-word representations directly contribute to the formation of target word representations. In contrast, encoder-decoder models largely relegate the task of capturing contextual dependencies to the decoder module. Through a series of complementary experiments, we identified two distinct roles performed by the decoder in encoder-decoder speech models when forming contextualized representations: attending to discrete prefix cues using self-attention, and attending to encoded audio from the encoder using cross-attention.

Our study validates the applicability of the adapted context mixing scores that were originally developed for text, and showcases their potential for discovering patterns of information flow in Transformer models of language in spoken modality. We hope this convergence in analytical methodologies between speech and text communities can pave the way for developing unified models for natural language processing (Chrupała, 2023). In line with previous research, our results discredit raw attention weights as a reliable source of information for information mixing in speech models.

# 7 Limitations

Our experiments rely on a specific phenomenon in French, as it provides a discrepancy between marking of number agreement in spoken versus written forms, and therefore provide us with an ideal case study for investigating context dependency and incorporating syntactic cues. However, in the case of homophony in French, there is one single cue word for each target word representation. In the future, we must search for other phenomena that allow us to investigate the role of multiple (converging or contrasting) sources of information when disambiguation is needed.

When choosing our target models, we have focused on models that are trained for automatic speech recognition. Models trained according to different objectives must also be considered, and the impact of pre-training vs. fine-tuning on context dependency and information mixing must be examined.

It is also worthwhile to use our adapted context-mixing methods to explore unsuspected phenomena in speech Transformers, without relying on strong prior hypotheses about model behavior.

## Acknowledgments

This publication is part of the project *InDeep: Interpreting Deep Learning Models for Text and Sound* (with project number NWA.1292.19.399) of National Research Agenda (NWA-ORC) programme. Funding by the Dutch Research Council (NWO) is gratefully acknowledged. We also thank SURF (www.surf.nl) for the support in using the National Supercomputer Snellius.

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

abs/2305.17733.

## A    Number homophony extraction procedure

Besides matching our defined templates based on
dependency trees, we considered a supplemental
set of conditions to ensure homophony exists in our
constructed data.

- Irregular nouns in the list [oeil, yeux, aïeul,
  aïeux, ciel, cieux, vieil, vieux] as well as
  nouns that end with *al* or *ail* are filtered out as
  target noun.

- Target verbs must be in the present or imper-
  fect tense, in any person-number except for
  1-plural and 2-plural.

- Target verbs in their different number, person,
  and tense must end with specific letters to en-
  sure they are pronounced the same in their dif-
  ferent forms. In the present tense, a verb must
  end with *e*, *es*, *e*, and *ent* in their 1-singular,
  2-singular, 3-singular, and 3-plural, respec-
  tively, while in the imperfect tense, the verb
  must end with *ais*, *ais*, *ait*, and *aient* in their
  1-singular, 2-singular, 3-singular, and 3-plural,
  respectively. To check for this condition we
  utilized inflecteur library[9] which is based on
  the DELA dictionary[10].

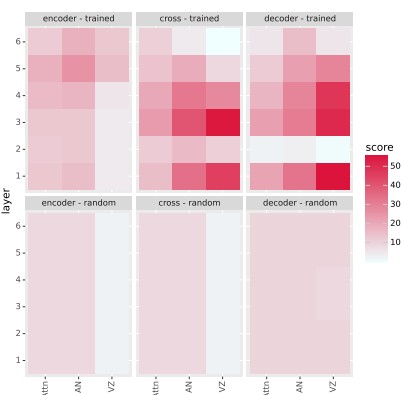

Figure A.1: Layer-wise cue contribution according to
different analysis methods averaged over all examples
for **Whisper-base**, trained (top) vs. randomly initialized
(bottom).

---

[9]https://github.com/Achuttarsing/inflecteur
[10]https://infolingu.univ-mlv.fr

## B    Cue contribution for other model sizes

The pattern of the results for layer-wise cue contri-
bution is the same among all model sizes and other
models within each family.

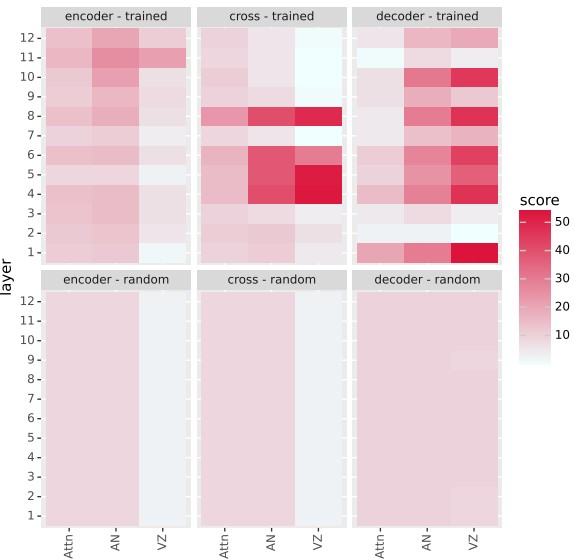

Figure B.1: Layer-wise cue contribution according to
different analysis methods averaged over all examples
for **Whisper-small**, trained (top) vs. randomly initial-
ized (bottom).

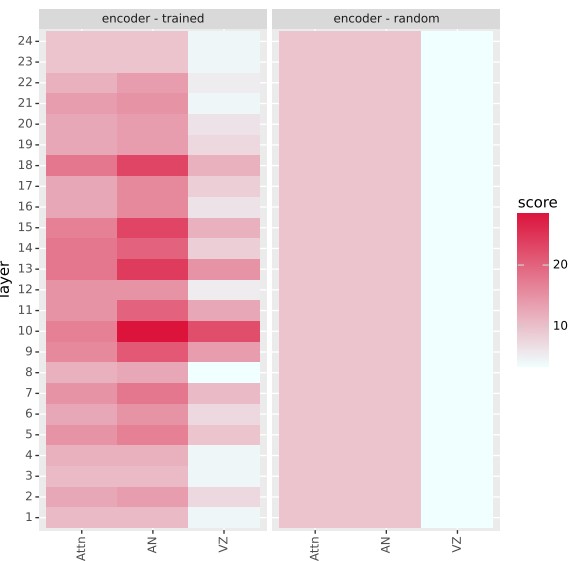

Figure B.2: Layer-wise cue contribution according
to different analysis methods averaged over all exam-
ples for **XLSR-1**, trained (left) vs. randomly initialized
(right).

# C  Example-wise visualization

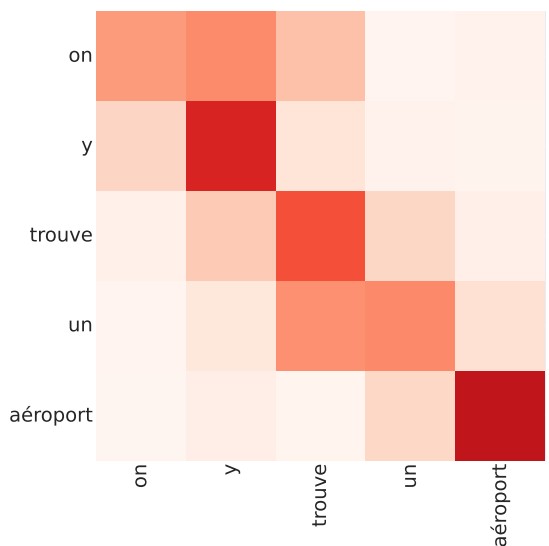

Figure C.1: **Within-encoder** Value Zeroing context mixing map for **XLSR-53** at layer 10 for our running example.

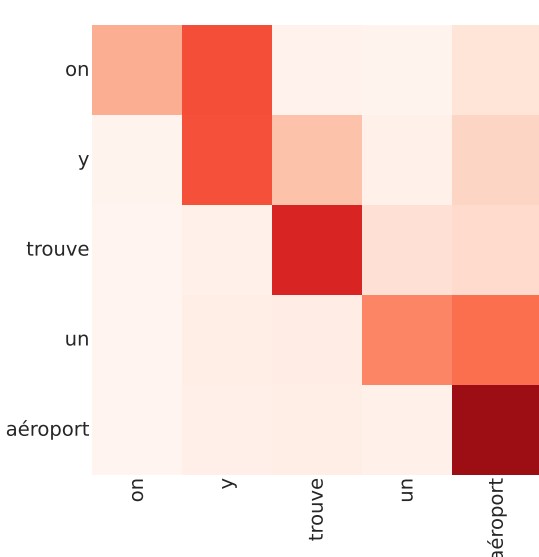

Figure C.3: **Within-encoder** Value Zeroing context mixing map for **Whisper-medium** at layer 10 for our running example.

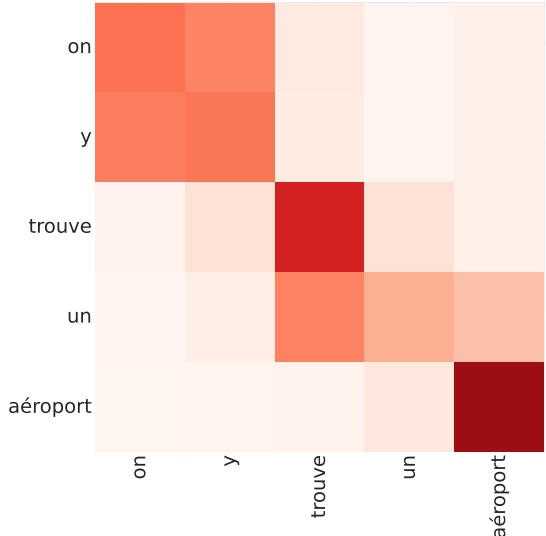

Figure C.2: **Within-encoder** Value Zeroing context mixing map for **XLSR-1** at layer 10 for our running example.

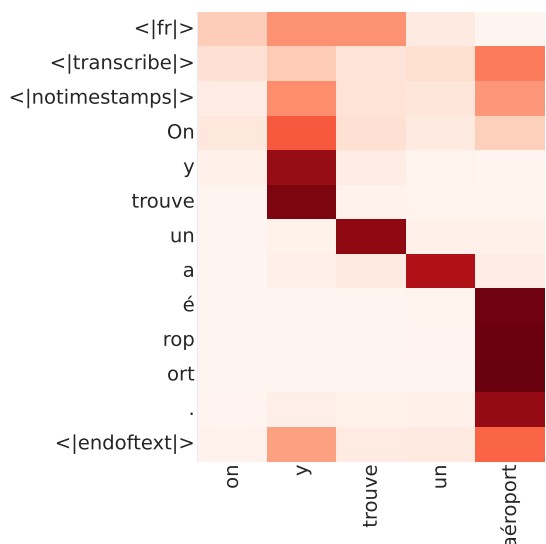

Figure C.4: **Cross** Value Zeroing context mixing map for **Whisper-medium** at layer 10 for our running example.