# OpenReview forum: "Homophone Disambiguation Reveals Patterns of Context Mixing in Speech Transformers"
_EMNLP/2023/Conference — EMNLP 2023 Main_

### Official Review · Reviewer_EiPu · 2023-08-10

**Soundness:** 3

**Excitement:**

4: Strong: This paper deepens the understanding of some phenomenon or lowers the barriers to an existing research direction.

**Paper Topic And Main Contributions:**

The paper is an interesting and novel study of how speech models learn syntactic dependencies and handle homophony. It also showcases the potential of context mixing methods for discovering hidden phenomena in speech Transformers.

**Questions For The Authors:**

How do you ensure that the context mixing methods are robust and reliable for speech models, especially when dealing with noisy or low-quality audio inputs?
How do you handle cases where there are multiple cues or conflicting cues for disambiguating a target word in speech?

**Reasons To Accept:**

It adapts state-of-the-art context mixing methods from the text domain to speech models and validates their applicability.
It reveals that encoder-only models rely on spoken cue-word representations to disambiguate target words, while encoder-decoder models mainly use text cue-word representations in the decoder prefix.

**Reasons To Reject:**

It only focuses on the encoder block of the Transformer architecture, and does not consider the decoder block or other components such as the output layer or the loss function. The authors acknowledge this limitation in their conclusion and suggest future work to extend their analysis to other parts of the model. Another possible weakness is that the paper relies on a specific linguistic phenomenon (homophony in French) to probe context mixing in speech models, which may not generalize to other languages or tasks.

**Reproducibility:**

4: Could mostly reproduce the results, but there may be some variation because of sample variance or minor variations in their interpretation of the protocol or method.

**Reviewer Confidence:**

3: Pretty sure, but there's a chance I missed something. Although I have a good feel for this area in general, I did not carefully check the paper's details, e.g., the math, experimental design, or novelty.

---

> ### Author Rebuttal · Authors · 2023-08-28
>
> **[Focusing on encoder block]**\
> We do not, in fact, focus just on the encoder block. Our experiments are conducted on both the encoder and decoder blocks of Transformers, considering every possible type of information interaction to mix the context: within-encoder, within-decoder, and cross-attention. However, analyzing context mixing in models with different training objectives, such as Clip or Diffusion models (which are based on Transformers as well but beyond the scope of this study), would be an interesting future direction.
>
> Copying answer to **[Single phenomenon]** for convenience:\
> Even though we opted for a single controlled task, we did not limit the analysis to a specific template and covered various types of syntactic templates in which homophony may appear.
> Additionally it is worth pointing out that there are many highly regarded studies which are limited to a single phenomenon in a single language, for example works investigating how language models capture long-distance subject-verb agreement. As long as that language is English, this narrow focus is usually not taken as a shortcoming. We would suggest that this type of paper is valuable, regardless of whether the language under consideration is English, French or any other language, if one makes a well-motivated choice (as we do) to study a particular phenomenon in that language, rather than trying out very many different languages and reporting only the results where the hypothesis holds (which we, emphatically, did not do).
>
> **[Q]**\
> Our templates are specifically designed to make sure that there is a single cue which determines the spelling of the target with no ambiguity. In our experiments we use clean speech so poor quality audio is unlikely to be a major factor in the results. Additionally we also make sure to analyze examples where the models generate the correct output transcription to make sure that the model is capturing the phenomena in question.

---

### Official Review · Reviewer_RLmt · 2023-08-10

**Typos Grammar Style And Presentation Improvements:** line 584
**Soundness:** 5

**Excitement:**

4: Strong: This paper deepens the understanding of some phenomenon or lowers the barriers to an existing research direction.

**Paper Topic And Main Contributions:**

The article describes a controlled analysis of the speech transformer model in an attempt to shed light on the mechanism by which such models build their representations. The analysis focuses on problems posed by homophony in French, and explores both state-of-the-art encoder-only systems and state-of-the-art encoder-decoder systems.

**Questions For The Authors:**

1. I think the example on 058-069 may contain an error. On 067, the authors state "here the singular determiner", but the determiner in their example on line 059 (_les_ livres) is plural.(Also, the small caps on Target and Cue on lines 060 and 067 do not appear to be used again in the text.)

2. Given the thoroughness of the surrounding description, the lack of mention/description of d_h in Equation (4) looks like an omission.

3. Does the fact that data are limited to only those cases in which all models produce an accurate transcription of Cue and Target(e.g. 395-396)  introduce a bias into the findings? It seems that the expectation voiced on lines 400-402 may not be obviously substantiated.

4. If the "score", "20", "10", etc textboxes at the right of Figures 1 and 2 are intended as some sort of legend, they are not onvious or sufficiently visible to be useful.

5. The example on 453-457 merits more explanation than the authors provide. The key points are likely to be missed by most readers.

**Reasons To Accept:**

Both the authors' findings and the method of obtaining them are interesting and compelling.

**Reasons To Reject:**

None. There are some minor points meriting comment or correction, see "Questions For The Authors" below.

**Reproducibility:**

4: Could mostly reproduce the results, but there may be some variation because of sample variance or minor variations in their interpretation of the protocol or method.

**Reviewer Confidence:**

4: Quite sure. I tried to check the important points carefully. It's unlikely, though conceivable, that I missed something that should affect my ratings.

---

> ### Author Rebuttal · Authors · 2023-08-28
>
> **[Q3]**\
> Evaluation-wise, we cannot expect a model to attend to the relevant part of the context of the label when it is unable to generate the true label. This is why we followed [1] and [2] and evaluated the model only on correctly classified examples. This choice does not pose a bias in our findings, as the performance of the models used in our study is impressively high for the task of ASR, and only a few samples have been filtered out.\
> [1]: https://arxiv.org/abs/2301.12971 \
> [2]: https://arxiv.org/abs/2304.14767
>
> **[Q1,2,4,5]**\
> We would like to thank you for pointing out the typos and presentation points.

---

### Official Review · Reviewer_AWuy · 2023-08-10

**Soundness:** 5

**Excitement:**

4: Strong: This paper deepens the understanding of some phenomenon or lowers the barriers to an existing research direction.

**Paper Topic And Main Contributions:**

The authors analyzed various context-mixing methods to see whether the existing speech transformer models look for syntactic cues for the correct transcription of homophones in French. The main contribution of the paper is toward explainable AI where we can analyze different transformer architectures in regard to the encoded speech representations. The results show that both encoder-only and encoder-decoder architectures attend to syntactic cues regarding the homophones in various layers in either the encoder layer of the former and the decoder or cross layers of the latter.

**Reasons To Accept:**

The topic of the paper is related to the conference. It is written clearly and there are no issues with the technical parts of the work. Overall the contribution of the paper is sound and highly recommend for acceptance.

**Reasons To Reject:**

There are no major limitations that would cause any risk of presentation.

**Reproducibility:**

5: Could easily reproduce the results.

**Reviewer Confidence:**

4: Quite sure. I tried to check the important points carefully. It's unlikely, though conceivable, that I missed something that should affect my ratings.

---

> ### Author Rebuttal · Authors · 2023-08-28
>
> Thank you for showing interest in our work. We kindly request that you elaborate on the specific aspects of our study that contributed to your score. Your score is greatly appreciated.

---

### Official Review · Reviewer_vPyo · 2023-08-11

**Soundness:** 5

**Excitement:**

4: Strong: This paper deepens the understanding of some phenomenon or lowers the barriers to an existing research direction.

**Missing References:**

Section 2 discusses all the related works in analyses of speech models and context-mixing separately, but it is not clear whether there has been prior work (with or without Attention Norms and Value Zeroing ) done to use context-mixing methods for analyses of speech models (especially the phenomenon of homophones).

**Paper Topic And Main Contributions:**

This paper focuses on using context-mixing methods built for the textual domain to investigate and analyze the influence of syntactic cues in disambiguating spoken words with similar pronunciation but different transcription in spoken language models. They use the phenomenon of homophony in the French to analyze the importance of syntactic cues in two types of transformer-based automatic speech recognition (ASR) models, (1) Encoder only where they experiment on models from Wav2vec2 family, and (2) Enoder-decoder models where they experiment on models from Whisper family. They extract the audio-transcription samples from the Common Voice Spoken corpus test set based on specific syntactic templates that satisfy the prior expectation of samples having homophony in French. Investigation of context-mixing on different transformer-based models (encoder only and encoder-decoder models) on these samples is done by visualizing per layer context-mixing scores for the cue words using state-of-the-art context-mixing methods namely, (1) Attention Norm, and (2) Value Zeroing. From their experiments, they were able to confirm the prior expectation of dependence of the target word transcription on their corresponding syntactic cue words. The main finding from the context-mixing scores for both the context-mixing methods was that in the encoder-decoder models, the syntactic dependency is captured primarily in the decoder part of the model. They further validated their claim by conducting experiments by training a simple classification model from the extracted target word representations for all the layers to predict the target grammatical number (binary classification with labels singular/plural), where the results implied that the middle layers primarily incorporated information to encode the number and syntactic cue representation. Their work establishes the applicability of the context-mixing methods developed for the textual domain to be used in the spoken domain for the interpretability of context influence and information flow.

**Questions For The Authors:**

Questions:

A. [Lines 305-310] Why do the authors just use the test set from the Common Voice Spoken corpus? Was the training set not available? Was it due to resource constraints (If so then it would be good to mention it).

B. [Lines 307-310] How exactly the dependency trees are used to discover instances of the templates mentioned in Lines 297-303?

C. [Lines 352-356] Is there a reason to use two different variants of the Wav2vec2 family? also, what is the reason to use one multi-lingual variant and one variant pre-trained on French speech?

D. [Lines 361-366] What is the reason for using Montreal Forced Aligner to extract start and end time, is does this give the best alignment results?

E. [Lines 419-420] Figure 1 illustrates layer-wise context-mixing score maps for just the XLSR-53 variant of the encoder-only model, and this statement is generalized for both the models, so does XLSR-1 give the same results? do both have high scores for the middle layer or is there some variance? This statement is premature/partially wrong to conclude if the paper does not provide evidence of both models having the same trends.

F. [Lines 438-443] Same as question (E), This statement is premature/partially wrong to conclude if the paper does not provide evidence of all model variants having the same trends.

G. [Lines 410-412] Why is random initialization the baseline of comparison of the layer-wise scores in Figure 1 \& 2? Why is the pre-trained weight initialization of these models considered baseline? It is not clear why random initialization is chosen as the baseline for comparison.

H. In Figure 3., it is not clear to me how only the cue word has a high context-mixing map score with the target word. From the context-mixing map, I can see that the cue word even has a high score with the token "alors" which is not part of the target word. Why is it so?

F. Section 2 discusses all the related works in analyses of speech models and context-mixing separately, but it is not clear whether there has been prior work (with or without Attention Norms and Value Zeroing ) done to use context-mixing methods for analyses of speech models (especially the phenomenon of homophones).

**Reasons To Accept:**

The paper investigates an interesting phenomenon of homophony in the spoken French language. It demonstrates the applicability of state-of-the-art context-mixing methods developed for the textual domain to study and interpret the influence of cue words in speech transcription tasks. Their study helps in concluding that cue words play an important role in properly transcribing spoken languages containing homophones (French in this work). The findings from the paper suggest that the decoder in the encode-decoder model plays a significant role in capturing syntactic dependencies, which has the potential use to train spoken language models in a parameter-efficient manner to tackle the problem of homophony in spoken languages. The layer-wise attention score visualizations provided in this work make it highly interpretable, and the cue contribution probing experiments validate their main findings.

**Reasons To Reject:**

Although the majority of concerns are addressed in the limitation section, the major concerns that I have are:

1. The study only focuses on a single phenomenon of Homophones and that too using a single language (French), which makes me question the applicability and generalizability of the text-based context-mixing methods for other spoken languages and phenomenons like polysemy (in speech transcription).

2. Although the authors state to release the code in the future, the paper does not have much information regarding the training hyperparameter (No section that discusses the training procedure of the models), like how long were the models trained and were these models trained till convergence to confidently conclude the final finding based on the provided layer-wise context-mixing score maps? Otherwise, it would be hard to trust these findings as there is no discussion on whether the results will remain the same irrespective of how long the model is trained.

3. The layer-wise context-mixing score analysis maps are only provided for single encoder-only and encoder-decoder model variants (Figure 1 \& Figure 2), which in my opinion makes it premature to conclude the prior hypothesis in a generalized statement (Line 419-420). If similar behavior exists for other variants then I believe their results should also be included in the paper to justify a generalized conclusion.

Overall although they have a descent analysis on using text-based context-mixing methods in the spoken language setting, I believe that more thorough justifications and evidence are needed to trust their claims.

**Reproducibility:**

4: Could mostly reproduce the results, but there may be some variation because of sample variance or minor variations in their interpretation of the protocol or method.

**Reviewer Confidence:**

4: Quite sure. I tried to check the important points carefully. It's unlikely, though conceivable, that I missed something that should affect my ratings.

**Typos Grammar Style And Presentation Improvements:**

[Lines 419-420, 438-443] Figure 1 \& 2 should be modified to include results for all the variants of both the family of models for proper comparison.

[Lines 419-420, 438-443] Figure 1 \& 2 should also include the context-mixing score maps for pre-trained weights along with the randomly initialized as well as fine-tuned weights.

Figure 3 should be modified with some sort of highlighting to better understand how the cue word has a high context-mixing map score with the target word.

In Table 1, it is unclear for a person who does not know the French language to understand the example properly, so it would be helpful to include the English translation in the examples.

---

> ### Author Rebuttal · Authors · 2023-08-28
>
> Many thanks for the detailed concerns you raised. We believe we can address the concerns, and hope to reassure you about the generalizability and strength of our findings.
>
>
> **[Single phenomenon]**\
> The reason for choosing homophony in the French spoken language was that we wanted to conduct controlled experiments to evaluate the applicability of context-mixing methods for speech Transformers. It provides a strong hypothesis regarding which part of the context is relevant when transcribing ambiguous target words. In contrast, tasks involving polysemy lack such decisive hypotheses due to the potential disambiguation of the target word through both semantic and syntactic cues. Even though we did not limit the analysis to a specific template and covered various types of syntactic templates in which homophony may appear.
>
> Additionally it is worth pointing out that there are many highly regarded studies which are limited to a single phenomenon in a single language, for example works investigating how language models capture long-distance subject-verb agreement. As long as that language is English, this narrow focus is usually not taken as a shortcoming. We would suggest that this type of paper is valuable, regardless of whether the language under consideration is English, French or any other language, if one makes a well-motivated choice (as we do) to study a particular phenomenon in that language, rather than trying out very many different languages and reporting only the results where the hypothesis holds (which we, emphatically, did not do).
>
> **[Training hyperparameter]**\
> We did not train the models ourselves (since training these pre-trained models is computationally very expensive). For the purpose of interpretability, we opted for off-the-shelf and state-of-the-art Transformer-based models. The training procedure for these models has been described in detail in Section 4.2. For specific training hyperparameters, please refer to the original papers cited in our work.
>
> In contrast to Whisper models, which have already been trained on the ASR task, the models in the Wav2vec2 family are pre-trained in a self-supervised manner. Therefore, they need to be fine-tuned for ASR. We used the commonly-used fine-tuned versions of these models from the HuggingFace hub, which are robustly and standardly fine-tuned, and have the highest download rate. For detailed fine-tuning hyperparameters, please refer to the footnotes 5 and 6 mentioned in our paper.
>
> **[Consistant results for other model sizes, and other models within a family in Figures 1 and 2]**\
> Figures 1 and 2 display the results only for XLSR-53 and Whisper-medium models which have 24 layers. In contrast to the other figures, it was difficult to integrate the results for all model sizes into one figure here. However, we assure the reviewer that the pattern of the results is the same among all models within each family. Our findings have been derived based on all the models used in the analysis. We will include the results for the other model sizes in the Appendix.
>
>
> **[QA]**\
> We only used the test split of the Common Voice corpus. This decision was made because the training and validation sets have already been utilized during the fine-tuning phase of models within the Wav2vec2 family. Therefore, the use of these sets would introduce bias and impede fair comparisons between models.
>
> **[QB]**\
> Using the SpaCy toolkit, it's possible to extract the dependency tree of an utterance and find the noun (verb) corresponding to a determiner (pronoun). After matching the template, we use the procedure described in Appendix A for homophony extraction. We refer the reviewer to the corresponding codes in the 'data_generator.py' file in our supplementary materials for more details.
>
> **[QC]**\
> We aimed to encompass a wide range of off-the-shelf speech Transformer models in our experiments to make our findings more generalized and comprehensive. For example, we observed the identical pattern of context mixing for both the multilingual and monolingual variants of Wav2vec2, validating the applicability of context mixing methods employed in our study for speech Transformers, regardless of the language of their training data.
>
> **[QD]**\
> The Montreal Forced Aligner consistently delivers more accurate results than available alternatives and its effectiveness has been demonstrated in previous works as well [1].\
> [1]: https://openreview.net/forum?id=AvcfxqRy4Y
>
> **[QE, QF]**\
> The results for XLSR-1 are the same as XLSR-53 (high scores in middle layers). Also, the results for other model sizes in the Whisper family are the same as for Whisper-medium (low scores in the encoder and high scores in the decoder). Therefore, we assure the reviewer that our observations are consistent across models within a family.
>
> **[QG]**\
> Non-trivial patterns can often be found in representations from randomly initialized models. These are due to systematic propagation of the input representation through the network, and are not due to any learning effect. In order to control for this phenomenon, it is crucial to use randomly initialized models as a reference point in interpretability studies.
>
> **[QH]**\
> Please note that the models are not specifically trained for homophony disambiguation. Therefore, the model can potentially encode different types of knowledge. Although it's not possible to draw conclusions based on only one example, the high-value contribution of token "Il" in the construction of token "alors" might be due to capturing other kinds of dependencies between these two words in the utterance.
>
>
> **[QF]**\
> To the best of our knowledge, this is the first work employing context-mixing methods for analyses of speech Transformers. Please let us know if you know of any previous efforts.

---

### Meta-Review · Area_Chair_hQS2 · 2023-09-18

**Recommendation:** 5

**Metareview:**

The paper investigates how syntactic cues affect the transcription task of homophones in speech models. It adapts various context-mixing methods from the text domain to study this phenomenon. The work primarily focuses on homophony in the French language, utilizing both encoder-only and encoder-decoder models for analysis.
Overall, the reviewers find the topic interesting and the paper well-written with clear technical aspects. The reviewers do not have major concerns but they concern about limited focus on a single phenomenon of homophones and that too using a single language (French).

---

### Decision · Program_Chairs · 2023-10-07

**Decision:**

Accept-Main

**Comment:**

The paper investigates how syntactic cues affect the transcription task of homophones in speech models. It adapts various context-mixing methods from the text domain to study this phenomenon. The work primarily focuses on homophony in the French language, utilizing both encoder-only and encoder-decoder models for analysis.
Overall, the reviewers find the topic interesting and the paper well-written with clear technical aspects. The reviewers do not have major concerns but they concern about limited focus on a single phenomenon of homophones and that too using a single language (French).